# Task-oriented Dialogue System for Automatic Disease Diagnosis via Hierarchical Reinforcement Learning

## Abstract

In this paper, we focus on automatic disease diagnosis with reinforcement learning (RL) methods in task-oriented dialogues setting. Different from conventional RL tasks, the action space for disease diagnosis (i.e., symptoms) is inevitably large, especially when the number of diseases increases. However, existing approaches to this problem typically works well in simple tasks but has significant challenges in complex scenarios. Inspired by the offline consultation process, we propose to integrate a hierarchical policy of two levels into the dialogue policy learning. The high level policy consists of a master model that is responsible for triggering a low level model, the low level policy consists of several symptom checkers and a disease classifier. Experimental results on both self-constructed real-world and synthetic datasets demonstrate that our hierarchical framework achieves higher accuracy and symptom recall in disease diagnosis compared with existing systems. Besides, both our datasets and codes will be released.

## 1 Introduction

With the development of electronic medical records (EMRs), researchers explore different machine learning approaches for automatic diagnosis (Shivade et al., 2013). Although impressive results have been reported for the identification of various diseases (Jonnalagadda et al., 2017; Doshi-Velez et al., 2014), they rely on well established EMRs which are labor-intensive to build. Moreover, each diagnosis requires EMR of the patient, while those systems lack the ability to obtain information through interaction with patients.

To relieve the pressure for constructing EMRs, researchers (Wei et al., 2018; Xu et al., 2019a) introduce task-oriented dialogue system to request symptoms automatically from patients for disease diagnosis. They formulate the task as Markov Decision Processes (MDPs) and employ reinforcement learning (RL) based methods in the policy learning for the agent. Existing frameworks utilize the setting of flat policy that treats diseases and all related symptoms equally. However, although RL-based approaches show positive results for symptom acquisition, when it comes to hundreds of diseases in real environment, flat policy is quite impractical.

In general, a particular disease is often related to a certain group of symptoms. As shown in figure 1, we present the correlation between diseases and symptoms, where x-axis represents symptoms and y-axis is the proportion of diseases related. From figure 1, there are obvious differences in the distribution of symptoms for different diseases. In offline consultation, doctors also do the pre-examination and triage according to the different symptoms that patient suffered, then doctors in different departments will make more detailed diagnosis. This method significantly reduces the workload of individual doctors and enables them to be more specialized in a certain field.

In the reinforcement learning algorithm, we found that Hierarchical Reinforcement Learning (HRL) (Parr & Russell, 1998; Sutton et al., 1999) conforms to this problem-solving logic. Recently, HRL, in which multiple layers of policies are trained to perform decision making, has been successfully applied to different scenarios, including course recommendation (Zhang et al., 2019), visual dialogue (Zhang et al., 2018), relation extraction (Feng et al., 2018), etc. The natural hierarchy of target tasks are modeled either manually or automatically, which motivates us to classify diseases

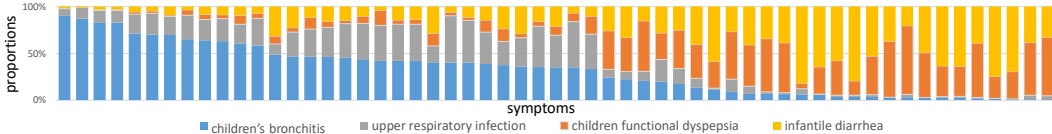

Figure 1: The disease distribution over symptoms in the real-world dataset (see section 3.1). x-axis stands for symptoms and y-axis is the proportion. Each bar describes the disease distribution given a symptom.

into different groups following the setting of departments in the hospital, and design a hierarchical structure for symptom acquisition and disease diagnosis.

In this paper, we classify diseases into several groups and build a dialogue system with a hierarchy of two levels for automatic disease diagnosis using HRL methods. The high level policy consists of a model named master and the low level policy consists of several workers and a disease classifier. The master is responsible for triggering a model in the low level. Each worker is responsible for inquiring symptoms related to a certain group of disease while disease classifier is responsible for making the final diagnosis based on information collected by workers. The proposed framework imitates a group of doctors from different departments to diagnose a patient together. Among them, each worker acts like a doctor from a specific department, while the master acts like a committee that appoints doctors to interact with this patient. When information collected from workers are sufficient, the master would activate a separate disease classifier to make the diagnosis. Models in the two levels are trained jointly for better disease diagnosis. We test our model in three large real-world datasets and a synthetic dataset for the evaluation of our model. Experimental results demonstrate that the performance of our hierarchical framework outperforms other state-of-the-art approaches.

We summary our contribution as follows: 1) We propose a new common-sensible method for automatic diagnosis. It simulates the real scene of clinical practice, and assigns patients to different workers through high-level policy, thereby reducing the action space and improving training efficiency. Also, the method can be compatible with different network models and training strategies; 2) We evaluate our method on three public datasets and a synthetic datasets to demonstrate the superiority of our framework with higher accuracy of disease and recall of symptoms; 3) We construct a detailed analysis of the effectiveness of each part of this method to confirm that it is reasonable and effective.

## 2 FRAMEWORK

In this section, we introduce our hierarchical reinforcement learning framework for disease diagnosis. We start with the hierarchical policy setting and then introduce every part of method. We further introduce reward shaping techniques which help us improve the model performance.

### 2.1 HIERARCHICAL POLICY

To reduce the large action space, we extend the RL formulation to a hierarchical structure with two-layer policies for automatic diagnosis. Following the *options* framework (Sutton et al., 1999), our framework is illustrated as in Figure 2(a). There are five components in our framework: master, workers, disease classifier, internal critic and user simulator.

Specifically, we divide all the diseases in $D$ into $h$ subsets $D_1, D_2, \ldots, D_h$, where $D_1 \cup D_2 \cup \cdots \cup D_h = D$ and $D_i \cap D_j = \emptyset$ for any $i \neq j$. Each $D_i$ is associated with a set of related symptoms $S_i \subseteq S$. While worker $w^i$ is responsible for collecting information about symptoms of $S_i$.

At turn $t$, The state $s_t$ will be encoded as one-hot vectors that reflects the status of each symptom, i.e., $s_t = [b_1^\top, b_2^\top, \cdots, b_{|S|}^\top]^\top$, and the master decides whether to collect symptom information (picking one worker to interact with user for several turns) or inform the diagnosis result (picking disease classifier to output the predicted disease). When agent inquires the symptom $i$, the patient will respond with *true/false/unknown* and agent will update the corresponding symptom statuses $b_i \in$

Figure 2: The training and inference process of our model. (a) The framework of our hierarchical reinforcement learning model with two-layer policies. (b) Illustration of the diagnosis process of our model with interactions between models in two levels. $w^i$ is the action invoking worker $w^i$ and $d$ is the action invoking disease classifier.

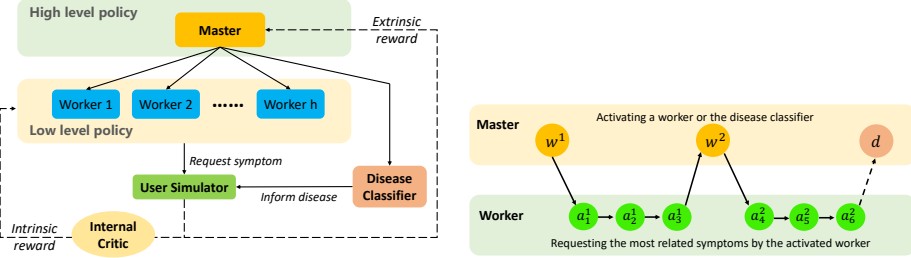

(a) Master-worker framework (training)      (b) Master-worker framework (inference)

$\mathbb{R}^3$. When agent informs a disease, the dialogue will be terminated as *success/fail* in terms of the correctness of diagnosis. Note that the not-requested symptoms are all encoded as $b = [0, 0, 0]$. After each action is executed, the agent will receive a reward $r$ based on the result of the action.

An illustration of the diagnosis process with interactions between models in two levels are presented in Figure 2(b). As for internal critic, it is responsible for both returning intrinsic reward to the worker and telling whether the subtask of the invoked worker is finished. In addition, the user simulator is applied to interact with our model and return extrinsic reward.

### 2.1.1 MASTER

The action space of master is $\mathcal{A}^m = \{w^i | i = 1, 2, \cdots, h\} \cup \{d\}$. The action $w^i$ indicates activating worker $w^i$ while $d$ is a primitive action which means activating the disease classifier. At each turn $t$, the master takes the dialogue state $s_t \in \mathcal{S}$ as input and takes an action $a_t^m \in \mathcal{A}^m$ according to it's policy. An extrinsic reward $r_t^e$ will be returned to master from the environment.

Once the master activates a worker, this worker will interact with user for $N$ turns until the subtask is terminated. Only after that master can take a new action and observe a new dialogue state. So, the learning problem of master can be formulated as a Semi-Markov Decision Process (SMDP), where the extrinsic rewards returned can be accumulated as the immediate rewards for the master (Ghavamzadeh, 2005). That is to say, after taking an action $a_t^m$, the reward $r_t^m$ for master can be defined as:

$$r_t^m = \begin{cases} \sum_{t'=1}^{N} \gamma^{t'} r_{t+t'}^e, & if \quad a_t^m = w^i \\ r_t^e, & if \quad a_t^m = d \end{cases} \tag{1}$$

Where $s'$ is the observed dialogue state of master after it takes an action $a^m$, $a^{m'}$ is the next action when the state is $s'$.

The objective of master is to maximize the extrinsic reward, thus we can write the master's loss function as follows:

$$\mathcal{L}(\theta_m) = \mathbb{E}_{s,a^m,r^m,s' \sim \mathcal{B}^m}[(y - Q_m(s, a^m; \theta_m))^2] \tag{2}$$

where $y = r^m + \gamma^N \max_{a^{m'}} Q_m(s', a^{m'}; \theta_m^-)$, $\theta_m$ is the network parameter at current iteration, $\theta_m^-$ is the network parameter of previous iteration and $\mathcal{B}^m$ is the fixed-length buffer of samples for master.

### 2.1.2 WORKER

The worker $w^i$ corresponds to the set of diseases $D_i$ and the set of relevant symptoms $S_i$, and the action space of worker $w^i$ is $S_i$. At turn $t$, if worker $w^i$ is invoked, the current state of master $s_t$ will be passed to worker $w^i$, then worker $w^i$ will extract the corresponding state representations $s_t^i$ from $s_t$ and take $s_t^i$ as input to generate an action $a_t^i \in \mathcal{A}_i^w$.

After taking action $a_t^i \in \mathcal{A}_i^w$, the dialogue is updated into $s_{t+1}$ and worker $w^i$ will receive an intrinsic reward $r_t^i$ from the module of internal critic. So the objective of worker is to maximize the expected cumulative discounted intrinsic rewards. The loss function of worker $w^i$ can be written as:

$$\mathcal{L}(\theta_w^i) = \mathbb{E}_{s^i, a^i, r^i, s^{i\prime} \sim \mathcal{B}_i^w}[(y_i - Q_w^i(s^i, a^i; \theta_w^i))^2] \tag{3}$$

where $y_i = r^i + \gamma_w \max_{a^{i\prime}} Q_w^i(s^{i\prime}, a^{i\prime}; \theta_w^{i-})$, $\gamma_w$ is the discounted factor of all the workers, $\theta_w^i$ is the network parameter at current iteration, $\theta_w^{i-}$ is the network parameter of previous iteration and $\mathcal{B}_i^w$ is the fixed-length buffer of samples for worker $w^i$.

### 2.1.3 DISEASE CLASSIFIER

Once the disease classifier is activated by master, it will take the master state $s_t$ as input and output a vector $\mathbf{p} \in \mathbb{R}^{|D|}$, which represents the probability distribution over all diseases. The disease with highest probability will be returned to the user as the diagnosis result. Two layers of Multi-Layered Perceptron (MLP) is utilized here for the disease diagnosis.

### 2.1.4 INTERNAL CRITIC

The internal critic is responsible for generating intrinsic reward $r_t^i$ to worker $w^i$ and judging the termination condition for the worker. In our task, a worker is terminated as failed when the number of subtask turns reaches $T^{sub}$, and terminated as successful when a correct symptom is requested by the agent. After that, a positive reward will be returned if the worker requests a correct symptom or disease. If there are repeated actions generated or the number of subtask turns reaches $T^{sub}$, $r_t^i$ would be negative. Otherwise, $r_t^i$ would be 0.

### 2.1.5 USER SIMULATOR

Following (Wei et al., 2018) and (Xu et al., 2019a), we set up a user simulator to interact with the agent. At the beginning of each dialogue session, the user simulator samples a user goal from the training set randomly. The simulator will initialize the dialogue session base on the explicit symptoms, and interacts with the agent based on the implicit symptoms. One dialogue session will be terminated as successful if the agent make the correct diagnosis, or failed if the informed disease is incorrect or the dialogue turn reaches the maximal turn $T$.

## 2.2 REWARD SHAPING

In practice, the number of symptoms a patient suffers from is much less than the size of symptom set $S$, which results in a sparse feature space. In other words, it is hard for the agent to locate the symptoms that the user truly suffers from. In order to encourage master to choose a worker that can discover more positive symptoms, we follow (Peng et al., 2018) and use the reward shaping method to add auxiliary reward to the original extrinsic reward while keeping the optimal reinforcement learning policy unchanged.

The auxiliary reward function from state $s_t$ to $s_{t+1}$ is defined as $f(s_t, s_{t+1}) = \gamma \lambda \phi(s_{t+1}) - \phi(s_t)$, $\phi(s)$ is the potential function, $\phi(s)$ counts the number of correct symptoms informed for a given state $s$, $\lambda > 0$ is a hyper-parameter which controls the magnitude of reward shaping and $\mathcal{S}_\perp$ is the terminal state set. Thus, the reward function for master will be changed into $R_t^\phi = r_t + f(s_t, s_{t+1})$.

## 2.3 TRAINING

Both the master policy $\pi^m$ and each worker's policy $\pi_i^w$ are parameterized via Deep Q-Network (Kulkarni et al., 2018; Mnih et al., 2015). In DQN, the action is often selected following an $\epsilon$-greedy policy. In our hierarchical framework, both the master and the workers behave following their own $\epsilon$-greedy policy for training and greedy policy for testing. During the training process, we store $(s_t, a_t^m, r_t^m, s_{t+N})$ in $\mathcal{B}^m$ and $(s_t^i, a_t^w, r_t^i, s_{t+1}^i)$ in $\mathcal{B}_i^w$. At each training step, we perform experience replay to update the current networks for both master and workers in $\mathcal{B}^m$ and $\mathcal{B}_i^w$ respectively, while the target networks are fixed during experience replay.

## 3 Experimental Datasets

We evaluate our framework on one simulated public datasets: **SymCat**, and three medical dialogue datasets: **MZ-4** (Wei et al., 2018), **MZ-10** and **Dxy** (Xu et al., 2019b). MZ-10 comes from our expansion on the basis of MZ-4, so that it can be close to a more complex dialogue environment in real medical dialogue.

### 3.1 Real-world Dataset

**MZ-4**    An existing dataset collected from real world for the evaluation of task-oriented DS (Wei et al., 2018). After proposed, it became one of the popular datasets for evaluating automatic medical systems. This dataset includes 4 diseases, 230 symptoms and 1733 user goals. Most of the existing RL-base methods have been tested on this dataset.

**MZ-10**    A newly constructed real-world dataset contains 4116 user goals that belong to 10 diseases, including typical diseases of digestive system, respiratory system and endocrine system. The raw data is collected from the pediatric department on a Chinese Online Healthcare Community[1]. Each user record consists of the self-report provided by the user and conversation text between the patient and a doctor. We hire experts with medical background to identify symptom expressions and label them with three tags ("True", "False" or "UNK") to indicate whether the user suffers this symptom. After that, experts manually link each symptom expression to a concept on SNOMED CT [2]. Symptoms extracted from self-report are treated as explicit symptoms and the ones extracted from conversation are implicit symptoms. At the same time, to ensure that the dataset is not too sparse, we only keep the symptoms with the top-100 frequency. The real-world dataset we constructed contains 4116 user goals, of which 80% for training ,10% for validating and 10% for testing.

**Dxy**    A Dialogue Medical dataset  (Xu et al., 2019b) contains data from a prevalent Chinese online healthcare website[3]. This dataset contains 527 user goals, including 5 diseases and 41 specific symptoms. Both datasets are already in the form of disease-symptoms pairs.

For real-world dataset, we compare the similarities of the counts of symptoms in each disease on the training set, and divide the diseases with higher similarity (above 0.5) into a group.

### 3.2 Synthetic Dataset

However, the number of disease and user goals in real-world dataset is still limited. In order to show the effectiveness of the HRL method, we build a synthetic dataset (SD) following (Kao et al., 2018) for further analysis of the method. It is constructed based on symptom-disease database called SymCat[4]. There are 801 diseases in the database and we classify them into 21 departments (groups) according to International Classification of Diseases (ICD-10-CM)[5]. We choose 9 representative departments from the database, each department contains top 10 diseases according to the occurrence rate in the Centers for Disease Control and Pre-vention (CDC) database.

In SymCat database, each disease is linked with a set of symptoms, where each symptom has a probability indicating how likely the symptom is identified for the disease. Based on the probability distribution, we generate record one by one for each target disease. Given a disease and its related symptoms, the generation of a user goal follows two steps. First, for each related symptom, we sample the symptom based on the probability. Second, a symptom is chosen randomly to be the explicit one (same as symptoms extracted from self-report in RD) and rest of true symptoms are treated as implicit ones. The synthetic dataset we constructed contains 30,000 user goals, including 90 diseases and 267 symptoms, of which 80% for training and 20% for testing.

Table 1 shows the details of all dataset used in this paper.

---

[1]http://muzhi.baidu.com

[2]https://www.snomed.org/snomed-ct

[3]https://dxy.com/

[4]www.symcat.com

[5]https://www.cdc.gov/nchs/icd/

Table 1: Overview of the Dataset.

| Name | ♯ of user goal | ♯ of diseases | ave. ♯ of im. sym. | ♯ of sym. |
|------|---------------|---------------|--------------------|-----------|
| MZ-4 | 1733 | 4 | 5.46 | 230 |
| MZ-10 | 3745 | 10 | 6.90 | 100 |
| Dxy | 527 | 5 | 1.67 | 41 |
| SymCat | 30,000 | 90 | 2.60 | 266 |

## 4 PERFORMANCE ON REAL-WORLD DATASETS

We train our model following the setting in Appendix A.1, and compare the effectiveness of different models in terms of accuracy, average turns and match rate.

### 4.1 MODELS FOR COMPARISON

We compare our model with some state-of-the-art reinforcement learning models for disease diagnosis.

- *Flat-DQN*: This is the agent of (Wei et al., 2018), which has one layer policy and an action space including both symptoms and diseases.

- *HRL-pretrained*: This is a hierarchical model from (Kao et al., 2018). The setting is similar to ours, however, the low level policy is pre-trained first and then the high level policy is trained. Besides, there is no disease classifier and the diagnosis is made by workers.

- *REFUEL*: This is a reinforcement learning method with reward shaping and feature rebuilding (Peng et al., 2018). It uses a branch to reconstruct the symptom vector to guide the policy gradient.

- *GAMP*: This is a GAN-based policy gradient network (Xia et al., 2020). It uses the GAN network to avoid generating randomized trials of symptom, and add mutual information to encourage the model to select the most discriminative symptoms.

- *KR-DS*: This is an improved method based on Flat-DQN (Xu et al., 2019b). It integrates a relational refinement branch and a knowledge-routed graph to strengthen the relationship between disease and symptoms.

- *FIT*: FIT is a novel non-RL framework for inquiring and diagnosing tasks (He et al., 2020). It propose a self-attention diagnosis model and use a multi-modal variational autoencoder (MVAE) as a probabilistic model dealing with partially observed data to improve the diagnostic accuracy.

### 4.2 OVERALL PERFORMANCE

Table 2 show the overall performance of different models respectively. In **MZ-4** dataset, most of the methods do not report the average turns and match rate of the dialogue, so we only compare the accuracy of the disease. Observed from **MZ-4** dataset, our method outperforms all the methods by at least 9%. In **MZ-10** dataset, the advantages of our model become more obvious. In comparison with flat-DQN, our method outperforms in both accuracy and match rates. When the number of symptoms increases, the symptom-disease and symptom-disease relationship becomes more complicated, which make Flat-DQN difficult to deal with. Our method makes the distribution of symptoms in each type of disease more logical by dividing diseases in different groups. So it can hit more symptoms during dialogue and provide more information for disease diagnosis.

Also, the results in **Dxy** dataset shows similar conclusions. Compared with other methods, our method can inquire more appropriate symptoms through interactions with patients, so as to make better decisions. In **Dxy** dataset, our methods outperforms other method in both accuracy and match rate by a larger magnitude, which means our method is more reasonable, efficient and logical.

Table 2: Overall performance on dataset. We conduct each experiment (Flat-DQN and HRLs) 5 times, and the reported number is the average, other numbers are copied from the corresponding literature, − denotes missing numbers, and Acc., M.R, Avg. T are the abbreviations of *Accuracy*, *Match Rate* and *Average Turns* respectively.

| | Dxy | | | MZ-4 | | | MZ-10 | | |
|---|---|---|---|---|---|---|---|---|---|
| Model | Acc. | M.R. | Avg. T | Acc. | M.R. | Avg. T | Acc. | M.R. | Avg. T |
| Flat-DQN | 0.731 | 0.110 | 3.92 | 0.681 | 0.062 | 2.53 | 0.408 | 0.047 | **19.51** |
| KR-DS | 0.740 | 0.267 | 3.36 | 0.730 | – | – | – | – | – |
| REFUEL | 0.757 | – | – | 0.718 | – | – | – | – | – |
| GAMP | 0.769 | 0.179 | 2.68 | 0.730 | – | – | – | – | – |
| FIT | 0.811 | – | – | 0.726 | – | – | – | – | – |
| HRL (w/o grouped) | 0.894 | 0.132 | 9.14 | 0.807 | 0.004 | 4.51 | 0.734 | 0.114 | 9.18 |
| HRL (w/o discriminator) | – | **0.512** | **16.84** | – | 0.120 | **12.93** | – | 0.330 | **17.49** |
| HRL (ours) | **0.848** | 0.426 | 14.97 | **0.820** | **0.127** | 10.16 | **0.747** | **0.363** | 17.37 |

### 4.3 ABLATION STUDIES

To verify the effects of the main components of our method, we further conducted a series of ablation studies. Because our HRL method consists of two parts, master-worker structure and disease classifier, we try not to divide symptoms into different group (Denoted as HRL (w/o grouped)) and remove the separate disease discriminator (Denoted as HRL (w/o discriminator)). Note that when we remove the master-worker structure and leave the work of judging the disease to the master, the model will degenerate into a simple Flat-DQN model.

The result is shown in Table 2. From the results, it can be seen that dividing diseases into different groups can effectively improve the recall of symptoms. Comparing HRL (w/o grouped) and Flat-dqn, a single disease classifier can greatly improve the efficiency of disease classification, while avoiding the dilemma caused by the uneven distribution of diseases and symptoms. Note that the implicit symptoms and the number of user goals in the Dxy dataset are limited, the advantages of the overall HRL method over its various parts are not very obvious.

## 5 PERFORMANCE ON SYNTHETIC LARGE-SCALE DISEASES

In current practice, most reinforcement learning methods are still powerless when faced with a large and sparse action space. However, the existing real-world datasets are limited in the number of diseases and the user goals. Therefore, in order to show the advantages and effects of our method on large-scale datasets, we will conduct validity experiments on Synthetic datasets.

### 5.1 OVERALL PERFORMANCE

As shown in Table 3, like Flat-DQN, most reinforcement learning methods face a difficult choice when facing this problem. Due to the large difference in the distribution of diseases and symptoms in the **SymCat** dataset, when the reward for the symptoms is too high, the agent will continue to inquiry symptoms without judging the disease. On the contrary, the agent will directly predict the disease and give up to search the sparse symptom space (like the Flat-DQN model we reported in table). Both of these situations are of limited help to real-world medical practice, but the balance between the two is difficult to hold and the system is extremely unstable.

Compared to baseline models, our method takes more turns to have interactions with the user so that it can collect more information of their implicit symptoms. With more information about implicit symptoms, our model significantly outperforms the other baselines in the diagnosis success rate, which shows our method is more reasonable and stable.

Table 3: Overall performance on SymCat dataset. We conduct all experiments 5 times, and the reported number is the average. − denotes missing numbers, and Acc., M.R, Avg. T are the abbreviations of *Accuracy*, *Match Rate* and *Average Turns* respectively.

| Model | Acc. | M.R. | Avg. T |
|---|---|---|---|
| Flat-DQN | 0.343 | 0.023 | 2.46 |
| HRL-pretrained | 0.452 | − | 6.84 |
| Ours | **0.504** | **0.495** | **12.96** |

## 5.2 STABILITY OF DISEASE CLASSIFIER

Because different diseases may have the same symptoms, it is an impossible task to distinguish all diseases correctly. At the same time, the same type of disease usually have similar treatment regimen. Therefore, if the disease classifier can make the judgment that is as close as possible to the correct disease, it can also help doctors roughly determine the type of disease. In order to have deeper analysis of the user goals which have been informed the wrong disease by the agent, we collect all the wrong informed user goals and present the error matrix in Fig 3.

It shows the disease prediction result for all the 9 groups. We can see the color of the diagonal square is darker than the others, which means the wrongly predicted disease and the correct disease are in the same group. This is reasonable because diseases in the same groups usually share similar symptoms and are therefore difficult to be distinguished. On the other hand, it also proved that even if the model cannot make correct predictions, it can still find right type of diseases to assist in real consultations.

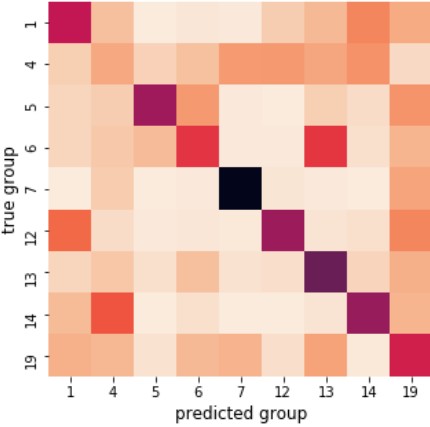

Figure 3: The error analysis for the disease classifier in different groups on our HRL model, the square with true group i and predicted group j means a disease in group i is misclassified into group j by the disease classifier, the darker the color, the higher the value.

## 5.3 EFFICIENCY OF DIFFERENT WORKERS

While proving the effect of the disease classifier, we also hope that the workers in each group can also play a positive role in the judgment of the model. We evaluate the performance of workers in terms of success rate, average intrinsic rewards and match rate. Match rate means the proportion of actions which have requested about the implicit symptoms that the user has. The results can be seen in Appendix A.2.We found most of the workers can successfully exit by querying the symptoms which patient suffered. It proves that workers in different groups can learn the symptom characteristics of the group, and use the knowledge to guide the consultation process.

## 6 RELATED WORK

This paper is related to two major research fields, namely hierarchical reinforcement learning and RL-based methods for disease diagnosis.

**Hierarchical reinforcement learning**   HRL has a hierarchical policy and has been proposed to solve the problems with large action space. One classic framework of HRL is *options* framework (Sutton et al., 1999), which involves abstractions over action space. At each step, the agent chooses either a one-step "primitive" action or a "multi-step" action (option). Kulkarni et al. (2018) proposed a hierarchical-DQN, which integrates deep Q-learning into HRL, to learn policies for different levels simultaneously. HRL has been successfully applied to different tasks and reached promising results. (Zhang et al., 2019; Wang et al., 2018; Zhang et al., 2018; Takanobu et al., 2018; Feng et al., 2018; Guo et al., 2018). Most existing works decompose the corresponding task into two steps manually, where the first step is finished by high level policy while the second step is finished by the low level policy. The task-specific design limits the generalization of HRL to complex tasks. Florensa et al. (2017) proposed a general framework that first learns useful skills (high level policies) in an environment and then leverages the acquired skills for learning faster in downstream tasks. What's more, some methods that generate or discover goals automatically when training the policies of two levels have been proposed. (Nachum et al., 2018; Florensa et al., 2018; Tang et al., 2018).

**RL-based methods for disease diagnosis**   There are some previous works that applies the flat RL-based method in the medical dialogue system Wei et al. (2018); Xu et al. (2019a); Tang et al. (2016); Kao et al. (2018); Peng et al. (2018) and generates positive results. In the work of Wei et al. (2018) and Xu et al. (2019a), both symptoms and diseases are treated as actions and a flat policy is used for choosing actions. After that, the HTC Research & Healthcare team did a lot of interesting work on this field. Kao et al. (2018) present a context-aware hierarchical reinforcement method, using policy gradients to make decisions based on the patient's age, gender, season, and explicit symptoms. Peng et al. (2018) proposed reward shaping and feature rebuilding techniques to help agent effectively learn a better strategy and Chen et al. (2019) introduced a new multiple action policy representation to help agent suggesting medical tests to facilitate disease diagnosis. While pursuing a higher accuracy of disease diagnosis, researchers also hope to make the inquiry process more efficient and logical. In order to encourage the agent to locate the symptoms more specifically, Xu et al. (2019a) introduce the knowledge graph into their dialogue system, Xia et al. apply GAN as policy network to capture the relations between different symptoms, these two works all reach a better accuracy than previous works.

## 7 CONCLUSIONS AND FUTURE WORK

In this work, we formulate disease diagnosis as a hierarchical policy learning problem, where symptom acquisition and disease diagnosis are assigned to different kinds of workers in the lower level of the hierarchy. A master model is designed in the higher level that is responsible for triggering models in low level. We improve the previous model by introducing disease classifier and internal critic and then train them together to reach the optimal. Besides the exist dataset, we extend the Muzhi dataset and build a synthetic dataset to evaluate our hierarchical model. To the best of our knowledge, this is the first time both kinds of datasets are used for model evaluation and we make both datasets public. The experimental results on all datasets demonstrate that our hierarchical model outperforms other RL-based model in both disease accuracy and symptom recall.

In the future, we would like to continue our research in three directions. First, we would like to make contribution on real-world dataset construction by introducing more diseases and annotate more samples. Second, although we can get a higher symptom recall, the symptom inquiry sequence is still disorderly. We can try to use some special reward setting to make it more logical. Third, we only use MLP as out deep learning network in this paper, and it would be interesting to integrate our hierarchical structure with other deep learning models and training strategies in the online self-diagnosis.

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

## A APPENDIX

### A.1 IMPLEMENTATION DETAILS

The $\epsilon$ for master and all the workers are all set to 0.1. For the master, the maximal dialogue turn $T$ is set to 22, it will receive a extrinsic reward of +1 if the master inform the right disease. Otherwise, it will receive a extrinsic reward of -1 if the dialogue turn reaches the maximal turn or a wrong disease is informed. At non-terminal turns, the extrinsic reward is 0. Moreover, the sum of the extrinsic rewards (after reward shaping) over one subtask taken by a worker will be the reward for master. The maximal dialogue turn $T^{sub}$ is set to 5 for each worker.

For master and all the workers, the neural network of DQN is a three-layer network with two dropout layers and the size of hidden layer is 512, learning rate for the DQN network is set to 0.0005. All parameters are set empirically and settings for the datasets are the same. In addition, all the workers are trained every 10 epochs separately during the training process of master. For the disease classifier, the neural network is a two-layer network with a dropout layer and the size of hidden layer is 512, learning rate for the network is set to 0.0005. Moreover, it's trained every epochs during the training process of master.

During the training process, it will take about 5000 epochs for the model to reach convergence, which takes about 18 hours given one GPU. For the best-performing model, $\gamma$ is set to 0.95, discounted factor $\gamma_w$ is set to 0.99, $\lambda$ in reward shaping is set to +1.

### A.2 THE PERFORMANCE OF DIFFERENT WORKERS IN SYMCAT DATASET.

| Group id | Success rate | Ave intrinsic reward | Match rate | Activation times |
|:---:|:---:|:---:|:---:|:---:|
| 1 | 48.6% | 0.031 | 16.74% | 0.615 |
| 4 | 54.6% | -0.150 | 5.02% | 0.375 |
| 5 | 38.8% | -0.013 | 7.96% | 3.252 |
| 6 | 48.0% | -0.036 | 9.58% | 0.942 |
| 7 | 48.3% | 0.057 | 18.57% | 1.280 |
| 12 | 43.0% | 0.021 | 11.26% | 0.666 |
| 13 | 52.4% | -0.138 | 7.18% | 0.823 |
| 14 | 72.2% | -0.111 | 3.77% | 0.614 |
| 19 | 47.4% | 0.031 | 22.72% | 1.124 |
| Average | 50.3% | -0.041 | 10.49% | 1.077 |

Table 4: The performance of different workers in in Symcat dataset.

