# OpenReview forum: "Task-oriented Dialogue System for Automatic Disease Diagnosis via Hierarchical Reinforcement Learning"
_ICLR.cc/2022/Conference — ICLR 2022 Submitted_

### Official Review · Reviewer_WycQ · 2021-10-26

**Correctness:** 3
**Technical Novelty And Significance:** 1
**Empirical Novelty And Significance:** 2
**Recommendation:** 3
**Confidence:** 3

**Main Review:**

Pros:
- This paper is well organized and easy to follow.
- The proposed method is well motivated and achieved strong empirical results in terms of disease classification accuracy and recall of symptoms.

Cons:
-  I am not very familiar with RL literature so I might be wrong. But to me, the proposed method looks like a straightforward application of HRL. Therefore, the technical contribution of the work is weak.
- Regarding the experimental results, I found that the proposed method requires significantly higher *Averaged Turns* for disease diagnosis compared to other baselines (as shown in Table 2). I don't understand why the authors bold the highest *Averaged Turns* in Table 2. But in task-oriented dialogues, accomplishing a user goal with fewer turns is better. Otherwise, the system could have checked all the possible symptoms one by one before making a final decision.


**Summary Of The Paper:**

This paper applies Hierarchical Reinforcement Learning (HRL) to automatic disease diagnosis in task-oriented dialogues setting. The authors argue that applying RL to automatic disease diagnosis is challenging because the action space (i.e., symptoms) is very large.
Therefore, they propose to learn a hierarchical dialogue policy where the high level policy is for categorizing patents into different groups and the low level policy is for checking symptoms and classifying diseases within a group.  Experimental results on multiple datasets show that the proposed HRL strategy achieve higher disease diagnosis accuracy compared to existing RL systems.






**Summary Of The Review:**

Overall, this is an OK work with limited novelty. I have some concerns about experimental results (see Main Review) which need to be clarified in the author's response.

---

### Official Review · Reviewer_65Z2 · 2021-10-26

**Correctness:** 3
**Technical Novelty And Significance:** 3
**Empirical Novelty And Significance:** 2
**Recommendation:** 5
**Confidence:** 3

**Details Of Ethics Concerns:**

Will the datasets contain the patient's personal information?

**Main Review:**

Strength:
1. The proposed HRL approach achieves better performance compared to the existing baselines.
2. New synthetic data was proposed for the task and can be useful for future study.


Weaknesses:
1. The idea of hierarchical RL is not particularly novel. It seems from Table 2, the existing best baseline is FIT, which is a non-RL approach, so it's not clear to me why RL is necessary to solve the automatic disease diagnosis. Since a non-RL approach achieves the best result, maybe we should develop on the non-RL approach?
2. It's worth showing some dialogue examples for the audience to understand what automatic disease diagnosis works under a task-oriented dialogue setting. It's not clear how the language generation part is handled in both simulated setting and real-life settings.
3. Many metrics are missing from the baselines to get a fair comparison in Table 2.


Questions:
1. Is the "disease discriminator" the same thing as "disease classifier"?
2. What does "user goal" mean under the task context? Are the users trying to get some treatment?
3. What exactly is the match rate of the dialogue? Is average turns the higher the better? It seems it's better if the diagnosis can be solved in reasonable amount of time instead of the longer the better?


Typo:
2.1.5, "base on" --> "based on"


**Summary Of The Paper:**

The paper tackles the problem of automatic disease diagnosis through reinforcement learning under a setting of task-oriented dialogues. The authors proposed to integrate hierarchical policies with two levels (one high-level master model, and one low-level policy) into the dialogue policy learning. Experiments on both real-life and synthetic data suggest the proposed approach is effective.

**Summary Of The Review:**

Overall I think the paper addresses an important problem, but it needs a bit more justification on why RL is necessary for this type of tasks, and some details of the paper can be clarified further.

---

### Official Review · Reviewer_J3rx · 2021-11-02

**Correctness:** 1
**Technical Novelty And Significance:** 1
**Empirical Novelty And Significance:** 2
**Recommendation:** 3
**Confidence:** 4

**Main Review:**


##########################################################################

Pros:

1. The paper takes one of the most important task, i.e., automatic disease diagnosis. However, the proposed approach only works at a semantic-level and is not an end-to-end approach. That is, it does not consider the NLU and NLg modules of the typical goal-oriented dialog systems and the paper is not clear about it. It took me a long time figure out what they are trying to do.

 2. Experiments are provided to show that the proposed methods work well. However, I am concerned about whether they are actually comparing against state-of-the-art systems?

3. The proposes a nice application of the hierarchical reinforcement learning.


##########################################################################

Cons:

1. The paper claims that in the conventional RL tasks, the action is not big, whereas the action space in disease diagnosis huge. However, there exists several works in RL where action space is huge, way bigger than the proposed work. Examples include generating natural language text, where action is equal to the vocabulary size. How does this work differ than those from the technical point of view.

2. Regarding comparison with the SOTA systems, I am familiar with at least one paper (there must be many out there) from EMNLP 2020. The title of the paper is: MedDialog: Large-scale Medical Dialogue Datasets.

3. From a technical point of view, I am not sure what is contribution of the paper.

4. In the experiments, the paper shows that having large number of turns as a positive sign, whereas in the goal-oriented dialog systems literature, the less number of turns in a conversation is better (as it is efficient to get job done quicker). However, the open-domain dialog systems literature is different where the goal is have longer conversations.

##########################################################################




**Summary Of The Paper:**

##########################################################################

Summary:
This works proposes dialogue policy learning by incorporating a hierarchical policy. The proposed hierarchical reinforcement learning-based approach has two models, i.e., one master model that instantiates low models. The low models have several symptoms checkers and disease classifier. Experiments are conducted on both real and synthetic datasets to demonstrate the efficacy of the proposed work.

##########################################################################

**Summary Of The Review:**

##########################################################################

Reasons for score:

Overall, I vote for rejection. I like the application side of the paper. But, technically, I am unable to see significant contribution. Moreover, the paper lacks in comparison with the state-of-the-art systems in the relevant domain.


##########################################################################

Questions for rebuttal:
Feel free to answer to all the concerns.

---

### Official Review · Reviewer_eBvJ · 2021-11-03

**Correctness:** 4
**Technical Novelty And Significance:** 2
**Empirical Novelty And Significance:** 2
**Recommendation:** 3
**Confidence:** 4

**Main Review:**

Strengths:
1. The task of automatic disease diagnosis that this work focuses on is worth exploring and has a very potential beneficial impact on society.
2. The paper is well organized and easy to understand

Weaknesses:
1. My main concern is the technical novelty of this work. This work is more like a combination of existing technologies. HRL, User Simulator, and Reward Shaping are all adopted in previous RL-based methods for disease diagnosis. The author's contribution on methodology is quite limited.

2. In section 4.3 ablation studies, more components (e.g., Reward Shaping) need to be considered to make the experimental results more convincing.

3. It would be helpful if the authors could provide some examples to understand how their systems behave.

**Summary Of The Paper:**

This paper introduces hierarchical reinforcement learning (HRL) into automatic disease diagnosis, which reduces the action space and improves training efficiency. Besides, the authors also expand an existing public dataset and build a synthetic dataset for evaluation. The Experimental results show that their proposed hierarchical framework achieves higher accuracy and symptom recall in disease diagnosis than existing several baselines.


**Summary Of The Review:**

This work focuses on a meaningful task and constructs datasets for better evaluation, but the limited contribution at the methodological level makes me inclined to think that it has not yet reached the bar for a top conference.

---

### Decision · Program_Chairs · 2022-01-20

**Decision:**

Reject

**Comment:**

The paper applies a reinforcement learning (RL) approach to a medical diagnosis dialog task. Motivated by a large action space, the authors utilize a hierarchical model where the higher level model triggers a lower level model comprising of symptom checkers and disease classifiers. They evaluate their approach on real-world and synthetic data sets.

Pros
+ The application (societal relevance) and the hierarchical approach (large action space) are motivated well
+ The paper is presented relatively clearly (with caveats: see reviewer comments) and improves performance over reasonable baselines (with caveats over one metric: why longer dialog is better?)

Cons
- The novelty of the work was not entirely clear, other than the application to a new task
- Lack of examples make it difficult to gauge the complexity of the task
- Ablation studies would also have provided better insight into task and the proposed model

The reviewers have several concerns about the work described in the paper. But the authors did not provide any response unfortunately.